# Genomic analysis unveils genome degradation events and gene flux in the emergence and persistence of *S*. Paratyphi A lineages

Jobin John Jacob[1◉], Agila K Pragasam[1◉], Karthick Vasudevan[1,2◉], Aravind Velmurugan[1], Monisha Priya Teekaraman[1], Tharani Priya Thirumoorthy[1], Pallab Ray[3], Madhu Gupta[3], Arti Kapil[4], Sulochana Putil Bai[5], Savitha Nagaraj[6], Karnika Saigal[7], Temsunaro Rongsen Chandola[8], Maria Thomas[9], Ashish Bavdekar[10], Sheena Evelyn Ebenezer[11], Jayanthi Shastri[12], Anuradha De[12], Shantha Dutta[13], Anna P. Alexander[14], Roshine Mary Koshy[15], Dasaratha R. Jinka[16], Ashita Singh[17], Sunil Kumar Srivastava[18], Shalini Anandan[1], Gordon Dougan[19], Jacob John[1], Gagandeep Kang[1], Balaji Veeraraghavan[1]*, Ankur Mutreja[19]*

1 Christian Medical College, Vellore, India, 2 REVA University, Bangalore, India, 3 Post Graduate Institute of Medical & Educational Research, Chandigarh, India, 4 All India Institute of Medical Sciences, New Delhi, India, 5 Kanchi Kamakoti Childs Trust Hospital, Chennai, India, 6 St. John's Medical College, Bengaluru, India, 7 Chacha Nehru Bal Chikitsalaya, Delhi, India, 8 Centre for Health Research & Development-Society for Applied Studies, New Delhi, India, 9 Christian Medical College, Ludhiana, India, 10 KEM Hospital & Research Centre, Pune, India, 11 The Duncan Hospital, Raxaul, India, 12 Topiwala National Medical College & BYL Nair Charitable Hospital, Mumbai, India, 13 ICMR-National Institute of Cholera and Enteric Diseases, Kolkata, India, 14 Lady Willingdon Hospital, Manali, India, 15 Makunda Christian Leprosy & General Hospital, Karimjang, India, 16 Rural Development Trust Hospital, Bathalapalli, Andhra Pradesh, India, 17 Chinchpada Christian Hospital, Nandurbar, India, 18 Department of Microbiology, SSN College, University of Delhi, Delhi, India, 19 Cambridge Institute of Therapeutic Immunology & Infectious Disease (CITIID), Department of Medicine, University of Cambridge, Cambridge, United Kingdom

◉ These authors contributed equally to this work.
* vbalaji@cmcvellore.ac.in (BV); am872@medschl.cam.ac.uk (AM)

**Data Availability Statement:** The authors confirm that all data are fully available without restriction. The whole genome sequenced raw read data is

## Abstract

Paratyphoid fever caused by *S*. Paratyphi A is endemic in parts of South Asia and Southeast Asia. The proportion of enteric fever cases caused by *S*. Paratyphi A has substantially increased, yet only limited data is available on the population structure and genetic diversity of this serovar. We examined the phylogenetic distribution and evolutionary trajectory of *S*. Paratyphi A isolates collected as part of the Indian enteric fever surveillance study "Surveillance of Enteric Fever in India (SEFI)." In the study period (2017–2020), *S*. Paratyphi A comprised 17.6% (441/2503) of total enteric fever cases in India, with the isolates highly susceptible to all the major antibiotics used for treatment except fluoroquinolones. Phylogenetic analysis clustered the global *S*. Paratyphi A collection into seven lineages (A-G), and the present study isolates were distributed in lineages A, C and F. Our analysis highlights that the genome degradation events and gene acquisitions or losses are key molecular events in the evolution of new *S*. Paratyphi A lineages/sub-lineages. A total of 10 hypothetically disrupted coding sequences (HDCS) or pseudogenes-forming mutations possibly associated with the emergence of lineages were identified. The pan-genome analysis

available at the European Nucleotide Archive (ENA) under study accession no. PRJEB44794 and PRJEB42715. The individual run accession numbers are listed in Supplementary Table S2.

**Funding:** This work was funded by grants from Bill & Melinda Gates Foundation, USA (Investment ID INV-009497 OPP1159351 to GK, BV and AM) for the Project "National Surveillance System for Enteric Fever in India. The funders had no role in study design, data collection and analysis, decision to publish, or preparation of the manuscript.

**Competing interests:** The authors have declared that no competing interests exist.

identified the insertion of P2/PSP3 phage and acquisition of IncX1 plasmid during the selection in 2.3.2/2.3.3 and 1.2.2 genotypes, respectively. We have identified six characteristic missense mutations associated with lipopolysaccharide (LPS) biosynthesis genes of *S.* Paratyphi A, however, these mutations confer only a low structural impact and possibly have minimal impact on vaccine effectiveness. Since *S.* Paratyphi A is human-restricted, high levels of genetic drift are not expected unless these bacteria transmit to naive hosts. However, public-health investigation and monitoring by means of genomic surveillance would be constantly needed to avoid *S.* Paratyphi A serovar becoming a public health threat similar to the *S.* Typhi of today.

## Author summary

*Salmonella enterica* serovar Paratyphi A remains one of the most common causes of enteric fever in the endemic regions of Asia. Within-host evolution of this human-restricted pathogen has led to the emergence of multiple lineages and sub-lineages through significant genomic rearrangement and pseudogene formation. Our study provides a snapshot overview of the gene flux and genome degradation events that shaped the evolution of *S.* Paratyphi A over the last 300 years. We leveraged the ongoing enteric fever surveillance program in India to collect 441 *S.* Paratyphi A clinical isolates, of which 152 were whole-genome sequenced. We carefully screened every internal node and cataloged mutations and gene flux at every stage of divergence/lineages throughout the phylogeny, revealing the role of key genetic changes in the evolutionary adaptation of *S.* Paratyphi A. Though several studies have previously reported the evolutionary trajectory of *S.* Paratyphi A, our study provides additional signatures with the most plausible evidence to date, backed by a molecular basis of adaptive mutations. The data from this study would merit further attention and interest in validating vaccines and developing novel diagnostics against *S.* Paratyphi A infections.

## Introduction

Enteric fever is a life-threatening systemic febrile illness caused by infections with *Salmonella enterica* serovar Typhi, Paratyphi A, B and C [1]. *S.* Typhi is the predominant cause of enteric fever, with an estimated 12–25 million cases of typhoid per year globally [2]. Among the three serovars that cause paratyphoid fever, *S.* Paratyphi A is the most prevalent and infections with *S.* Paratyphi B and C serotypes are extremely rare [3]. Both Typhoid and paratyphoid infections are endemic in parts of South Asia and South East Asia [4]. Though only limited data is available on the true burden of *S.* Paratyphi A in these regions, it is estimated to cause around 5 million cases of enteric fever annually [5]. However, the actual number of infections was underestimated as paratyphoid is clinically indistinguishable from typhoid fever [6]. Recent data suggests that the proportion of enteric fever cases caused by *S.* Paratyphi A appears to have significant regional variability in the endemic regions of South Asia. Notably, the contribution of paratyphoid was estimated to be 14.5–37% in population-based studies, while hospital-based studies estimate 10.3–54.5% of cases of the total enteric fever cases [7].

The sequential emergence of antimicrobial resistance in serovar Typhi over the past 50 years is well documented. Clinical, laboratory and genomic features of the evolution of antimicrobial resistance in *S.* Typhi against chloramphenicol (1960), first-line antimicrobials (1990),

fluoroquinolones, third-generation cephalosporins and azithromycin are already established [8,9]. However, unlike *S.* Typhi, serovar Paratyphi A is predominantly susceptible to most antibiotics, including the traditional first-line agents (ampicillin, chloramphenicol and co-trimoxazole). Nevertheless, high fluoroquinolone non-susceptibility (FQNS) in *S.* Paratyphi A has been witnessed in recent years, with sporadic reports of multidrug-resistant (MDR) and azithromycin-resistant isolates [10,11].

*S.* Paratyphi A was found to have substantial regional differences with the emergence of seven distinct lineages (A-G), each having originated in a specific geographical location [12]. Among the lineages, A and C have expanded throughout South Asia and Southeast Asian countries to become successful clones, whereas other lineages are still rare. Unlike *S.* Typhi, the genome-level difference of *S.* Paratyphi A was investigated in only a few isolates [13,14]. Interestingly, evolutionary changes in *S.* Paratyphi A by means of gene gain or loss or mutations are mostly considered transient and are continuously removed by purifying selection [12]. However, the diversification and expansion of these lineages were not correlated with mutations previously.

We have performed whole genome sequencing (WGS) within the existing national enteric fever surveillance system named Surveillance of Enteric Fever in India (SEFI). SEFI was established as a 3-tiered surveillance system with 19 different participating sites, consisting of community-level healthcare settings (Tier 1), secondary hospitals (Tier 2) and tertiary care hospitals (Tier 3). The Tier 1 surveillance sites consist of 1 rural and three urban sites representing different geographic settings and population densities. Tier 2 sites (5 rural and one urban) represent a predefined geographic catchment population, while tier 3 sites represent a diverse population with no distinct catchment areas, as this was culture-confirmed laboratory-based surveillance conducted at nine large tertiary care centers. Here, we examined the phylogenetic distribution of *S.* Paratyphi A isolates collected as part of SEFI. We also examined the gain, loss and inactivation of genes at the genomic level to shed light on the ongoing process of evolution in *S.* Paratyphi A.

## Results

### Surveillance of *S.* Paratyphi A infections

During the study period between October 2017 to September 2020, 441 *S.* Paratyphi A were isolated from blood and bone marrow cultures performed at all study sites. Laboratory-based surveillance in tertiary care hospitals yielded significant positivity rates of up to 80.27% (*n = 354/441;* 95% CI[76.25–83.89%]), followed by 12.24% (*n = 54/441;* 95% CI[9.33–15.67%]) in secondary care hospitals and 7.48% (*n = 33/441; 95% CI[5.21–10.35%])* from community cohorts. The isolation rates of *S.* Paratyphi A were four to eight times less compared with *S.* Typhi across various Tiers, as described in **S3 Table**. Overall, *S.* Paratyphi A comprised 17.6% (*441/2503;* 95% CI[16.14–19.17%]) of total enteric fever cases in India and was majorly recorded in tertiary care settings.

### Antimicrobial susceptibility testing of *S.* Paratyphi A isolates

The antimicrobial susceptibility test demonstrated that none of the tested *S.* Paratyphi A clinical isolates (*n = 441*) was MDR as all isolates were susceptible to each of the traditional first-line antibiotics (ampicillin, chloramphenicol, and co-trimoxazole). Fluoroquinolone non-susceptibility was nearly 98.9%, while a high degree of susceptibility to current alternative treatment options was recorded (100% susceptibility to azithromycin and ceftriaxone) (**S4 Table**). Overall, Indian *S.* Paratyphi A isolates were found to be generally non-susceptible to ciprofloxacin (*n = 436/441*), while they continue to be susceptible to first-line agents.

## Phylogeny and Population structure of *S*. Paratyphi A

Phylogenetic relationship of 552 *S*. Paratyphi A isolates based on 4,458 core genome SNPs showed the distribution of study isolates within a global genomic framework. The observed global phylogeny clustered the isolates into seven previously defined lineages (A-G), in which the study isolates were distributed between lineage A (65.8%; 100/152), C (26.3%; 40/152) and F (7.9%; 12/152) (**Fig 1**). Isolates belonging to lineage A were grouped into sequence type 129 (ST129), while lineages B-F were predominantly ST85. The single isolate clustered in lineage G was distinct and belonged to ST479, which varied two-locus from ST85. RhierBAPS (level 1) yielded five clusters, while level-2 clustering has distinguished a total of 21 sub-lineages (**S2 Table**). Though previous studies have described the sub-lineage level distribution of *S*. Paratyphi A isolates, we have used the recently developed 'Paratype scheme' [15] to define sublineages/genotypes within lineage A, C and F. We identified nine genotypes (2.4.1–2.4.9) within the dominant lineage A (genotype 2.4) based on Paratype scheme. Geographical distribution of lineage A isolates showed genotype 2.4.3 (previously A1) being predominant in Nepal, 2.4.1 (formerly A2) was present in both Nepal and India and 2.4.4 (previously A3) was primarily found in Bangladesh. Genotype 2.4.2 was predominantly seen in India with a sparse presence in other South Asian countries. The Paratype scheme assigned five new genotypes (2.4.5– 2.4.9), mainly consisting of Indian isolates. Among the new genotypes, 2.4.5 have been circulating globally, while 2.4.6, 2.4.7 and 2.4.8 consist of Indian isolates distributed distinctly in different geographic regions across the country. Notably, genotype 2.4.9 was geographically confined to a single site in Northern India, indicating a large localized outbreak. The geographical distribution of *S*. Paratyphi A genotypes from the study collection is shown as a scattered pie chart (**S1 Fig**).

The existing population structure defining sub-lineages of C (C1-C5) was not consistent with RhierBAPS clusters due to its genomic diversity, unlike regionally restricted lineage A. Sub-lineages C1 and C2 were represented by polytomies, while C4 and C5 were not following the BAPS level 2 clustering (**S2 Table**). The classification of lineage C (2.3) based on the Paratype scheme provides genotypes 2.3.1 (previously C5), 2.3.2 and 2.3.3 (formerly C4). The geographical distribution of global *S*. Paratyphi A isolates showed genotype 2.3 (previously C3) was represented by isolates originating from Africa and Pakistan. Genotype 2.3.2 were isolates predominantly from south Asia, while genotype 2.3.3 isolates were mainly from China, Southeast Asia, and South Asia. Similarly, the first cluster in sub-lineage C5 was designated as genotype 2.3.4 with isolates almost exclusively from India (80%; 20/25), whereas the second cluster (referred to as 2.3.1) was represented by outbreak isolates from Cambodia (**S2 Fig**). Genotyping of lineage F (genotype 1) was predicted to contain four sub-clusters (1, 1.1, 1.2.1 and 1.2.2), of which 1.2.2 comprised contemporary *S*. Paratyphi A isolates from both India (the present study isolates) and Bangladesh.

## Quinolone resistance determining mutations and plasmids

Reduced susceptibility to fluoroquinolones in dominant lineages (A, C and F) of *S*. Paratyphi A was driven mainly by *gyrA*-S83F substitutions, with a few isolates harbouring *gyrA*-S83Y (predominantly genotype 2.3.4) variant. Also, some isolates belonging to genotype 2.3.1 were fluoroquinolone-susceptible (**Fig 1**). Plasmid profiling revealed that most of the lineage C isolates (*n = 116*) harboured a ColRNAI plasmid with no AMR genes. Interestingly, isolates belonging to genotype 1.2.2 (*n = 27*) possessed IncX1 plasmid, while the MDR isolates from the global collection carried the AMR genes in either IncFIB or IncH1B plasmid.

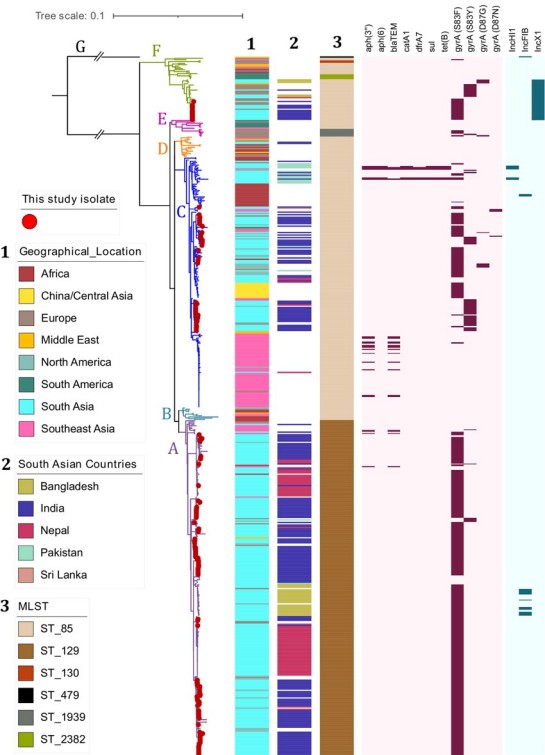

**Fig 1.** *Phylogenetic distribution of contemporary Indian S. Paratyphi A isolates in a global context*: **Rooted maximum likelihood phylogenetic tree of contemporary Indian S. Paratyphi A (*n = 152*), combined with global genome collection (*n = 400*) representing the current global distribution.** The tree was derived from 4286 SNPs mapped against the reference genome of *S.* Paratyphi ATCC 9150 (Accession No: CP000026.1) using Snippy and rooted to the outgroup strain (ERR028986: Lineage G). Red-colored dots at the tip of the branches indicates the position of this study isolates. Contemporary Indian *S.* Paratyphi A isolates of this study were found distributed across the global tree with both lineages A, C and F. Genomes with their respective metadata are labeled as color strips and key for each variable were mentioned. Strip 1 and 2 indicate the location and 3 represent MLST of each isolate. Heatmap represents the QRDR mutations that confer resistance to fluoroquinolone and the presence of plasmids. The scale bar indicates substitutions per site. Color keys for all the variables are given in the inset legend. The tree was visualized and labeled using iTOL (https://itol.embl.de/).

## Lineage-specific evolution of *S.* Paratyphi A

Mutations and gene flux that defines or drives the lineages or sub-lineages of *S.* Paratyphi A were identified from the population structure. The role of gene flux in evolution was determined by pan-genome analysis, while gene inactivation (frameshift mutations) and non-synonymous substitutions were determined by accessing the variant type. Synonymous mutations were not considered as their effect on evolution is likely negligible on the short evolutionary timescale captured in modern molecular epidemiological studies.

Pan-genome analysis revealed the variation of gene content between *S.* Paratyphi A genomes. About 73.8% (3944/5344) were considered core genes (found in >99% genomes),

**Table 1. Loss and Gain detected between phylogenetic lineages/genotypes of *S.* Paratyphi A.**

| S. No | Gene/ Region | Lineage/Genotype | Remarks |
|---|---|---|---|
| 1 | SPI-2 | A-F | Either lost in G or gained by A-F |
| 2 | P2/ PSP3 like phage | 2.3.2/2.3.3 | Gained by 2.3.2/2.3.3 ($C_4$) |
| 3 | IncX1 plasmid | 1.2.2 | Gained by 1.2.2 |

while 18.7% (997) genes were shared by ≤15% of isolates among the 552 screened (S3 Fig). Lineage-specific gene gain or loss during the evolutionary process showed the phylogenetically distinct lineage G lack SPI2. Given that lineage G is an outgroup and there is only a single isolate belonging to the lineage, this observation is insufficient for any evolutionary inference (Table 1). Gene gains that likely represent the host adaptation or pathogenicity with respect to the phylogenetic lineages were rather limited to mobile genetic elements. For example, the C4 sub-lineage (genotype 2.3.2 and 2.3.3) of *S.* Paratyphi A has acquired prophage regions P2/ PSP3 phage (S4 Fig). Interestingly, genotype 1.2.2 was found to have acquired IncX1 plasmid similar to pK43 previously reported from *S.* Typhi. (CP029854.1). The acquisition of IncX1 would be the most likely event that resulted in the emergence of 1.2.2 from the previously sporadically reported Lineage F (Genotype 1) (Fig 1).

Accumulation of HDCS/genome degradation events during the evolution provides insights into the continuous host adaptation or adaptive selection of *S.* Paratyphi A. We identified several lineage-specific HDCS since they diverged from ancestral lineages. In addition to the 133 pseudogenes conserved across all lineages except in lineage G, 50 additional genes were identified to be associated with loss of gene function through nonsense substitutions or frameshift mutations (S5 and S6 Tables). A total of 10 HDCS or pseudogene-forming mutations that could be associated with the emergence of lineages are listed in Table 2. Gene flux information and HDCS specific to lineages during the evolution of *S.* Paratyphi A are overlaid on a timed phylogenetic tree generated using Figtree in Fig 2.

Time-scaled Bayesian phylogenetic analysis showed that the model combination best fitted the data was a relaxed molecular clock paired with constant population size. This analysis dated the most recent common ancestor (MRCA) of *S.* Paratyphi A to the year 1693 (95% HPD 1540–1799) when a single isolate belonging to the distinct clade (lineage G) was excluded for bayesian analysis. The dominant lineages C and A have likely diversified between 1835 (95% HPD: 1804–1873) and 1856 (95% HPD: 1833–1884), respectively. Similarly, genotype 1.2.2 is estimated to have expanded in the year 1877 (95% HPD: 1844–1901) by acquiring the IncX1 plasmid. Overall, the estimated evolutionary rate was 4.008 x $10^{-5}$ or 0.301 nucleotide substitutions/site/year (s/s/y).

### Mutations in O-antigen biosynthesis genes

Mutation analysis of O-antigen biosynthesis genes (*rfb* region) showed the region carrying six characteristic missense mutations in comparison with the ATCC 9150 reference strain (vaccine candidate). Mutations in *rfb* gene cluster consist of single amino acid substitutions in *rfbG* (H348R), *rfbD* (G262S), *rfbE* (S167L), *rfbS* (C249S), *rfbB* (H176Y) and *rfbC* (E154K). Interestingly, these mutations are associated with lineages/genotypes currently circulating in south Asian countries (S5 Fig). For instance, genotypes carrying a characteristic missense mutation in LPS O-antigen biosyntheses such as 1.2.2 (*rfbC*: E154K), 2.3.3 (*rfbS*: C249S) and 2.4.2 (*rfbD*: G262S) are increasingly being detected, particularly in India. The predicted free energy gap difference (ΔΔG) between the wild type and mutant protein measures how the mutation impacts the protein stability. The ΔΔG values of different *rfb* gene mutations indicated stabilizing scores except for *rfbC*: E154K (ΔΔG = -4.06; S7 Table). However, the impact of these mutations in terms of vaccine efficacy is yet to be studied.

### Discussion

Genome analysis of 152 *S.* Paratyphi A isolates collected from different geographical locations in India between 2017 and 2020 revealed evolutionary changes that led to lineage diversification as well as its role in the persistence and spread of the bacterium. Comparative genome

**Table 2.  List of functional gene inactivation mutations identified between phylogenetic lineages.**

| S. No | Gene | Locus tag | Mutation | Lineage/Genotype | Function/Remarks |
|-------|------|-----------|----------|------------------|------------------|
| 1 | *tinR* | SPA2451 | Ile51fs | F/1 | Lrp/AsnC family transcriptional regulator (Toxin repressor) |
| 2 | *bcfB* | SPA0022 | Asn4fs | F/1 | Fimbrial biogenesis chaperone BcfB |
| 3 | - | SPA2644 | Asp60fs | E | Membrane transporter TctB family protein |
| 4 | *uhpB* | SPA3639 | Ile167fs | A-E | Signal transduction histidine-protein |
| 5 | - | SPA3466 | Ala642fs | A-E | AsmA family protein |
| 6 | *garD* | SPA3119 | Lys132fs | A-E | Galactarate dehydratase |
| 7 | - | SPA0042 | Ile438fs | A & B/2.4 | Glycoside hydrolase family 31 protein (disrupts biofilm formation) |
| 8 | - | SPA0505 | Pro305fs | A | Amino acid permease |
| 9 | *tdcD* | SPA3111 | Tyr163fs | $A_1$ /2.4.3 | Propionate kinase |
| 10 | *ompS1* | SPA0875 | Asn115fs | $C_5$/2.3.1 | Unknown function in virulence and biofilm formation |

fs: Frameshift

analysis unambiguously placed the contemporary *S.* Paratyphi A isolates from India into three lineages, with lineages A and C being dominant. This concurs with the previous analysis that reported the placement of present-day south Asian isolates in these three lineages [9,12,16]. In addition to the the current classification of sub-lineages within lineages A, C and F, a recently developed Paratype genotyping scheme [15] has further improved the sub-lineage level classification of major lineages. Our results provide a more comprehensive insight into the population structure and geographical distribution of *S.* Paratyphi A isolates in south Asian countries, particularly in India. Overall, the contemporary Indian *S.* Paratyphi A isolates clustered closely with those originating from Bangladesh, Nepal and Pakistan, suggesting a regional circulation of these lineages across south Asia.

Geographical distribution of genotypes provided a cross-sectional view of the *S.* Paratyphi A population in which certain genotypes are geographically confined to particular countries. Accordingly, the sub-clusters of lineage A are dominated by isolates from Nepal (2.4.3 & 2.4.1), Bangladesh (2.4.4), and India (2.4.2). Within lineage C, genotype 2.3 predominantly contains isolates from Africa and Pakistan. Similarly, isolates from India (2.3.2), China (2.3.3) and Cambodia (2.3.1) were distributed as geographically confined sub-lineages [17,18]. The phylogenetic positioning of contemporary *S.* Paratyphi A isolates in lineage F was unexpected; however, recent reports from Bangladesh also documented similar findings [15,16]. A closer look at the lineage F isolates revealed the positioning of older isolates from the global collection in genotype 1/1.1, while the contemporary isolates from India and Bangladesh form genotype 1.2.2. The emergence of genotype 1.2.2 can be attributed to the acquisition of the IncX1 plasmid, highlighting the role of horizontal gene transfer in favouring the successful evolution and long-term persistence of these lineages.

Antimicrobial resistance determined by phenotypic and genomic analysis of the study isolates showed low-level resistance to antimicrobials except for fluoroquinolones. These results were consistent with the previous estimates as most of the studies from south Asia report either no or low levels of multidrug resistance [10]. Though MDR phenotypes were observed in a few *S.* Paratyphi A strains from the global collection, the plasmid was eventually lost during the evolution due to the greater fitness of antibiotic-sensitive strains [12]. On the contrary, fluoroquinolone non-susceptibility (FQNS) was high amongst *S.* Paratyphi A in South Asia, with FQNS strains from the SEFI collection accounting for 98.9% of all isolates [19].

The FQNS *S.* Paratyphi A were predominantly single QRDR mutants (*gyrA*-S83F) and distributed across the dominant phylogenetic lineages (A, C and F). Interestingly, the success of

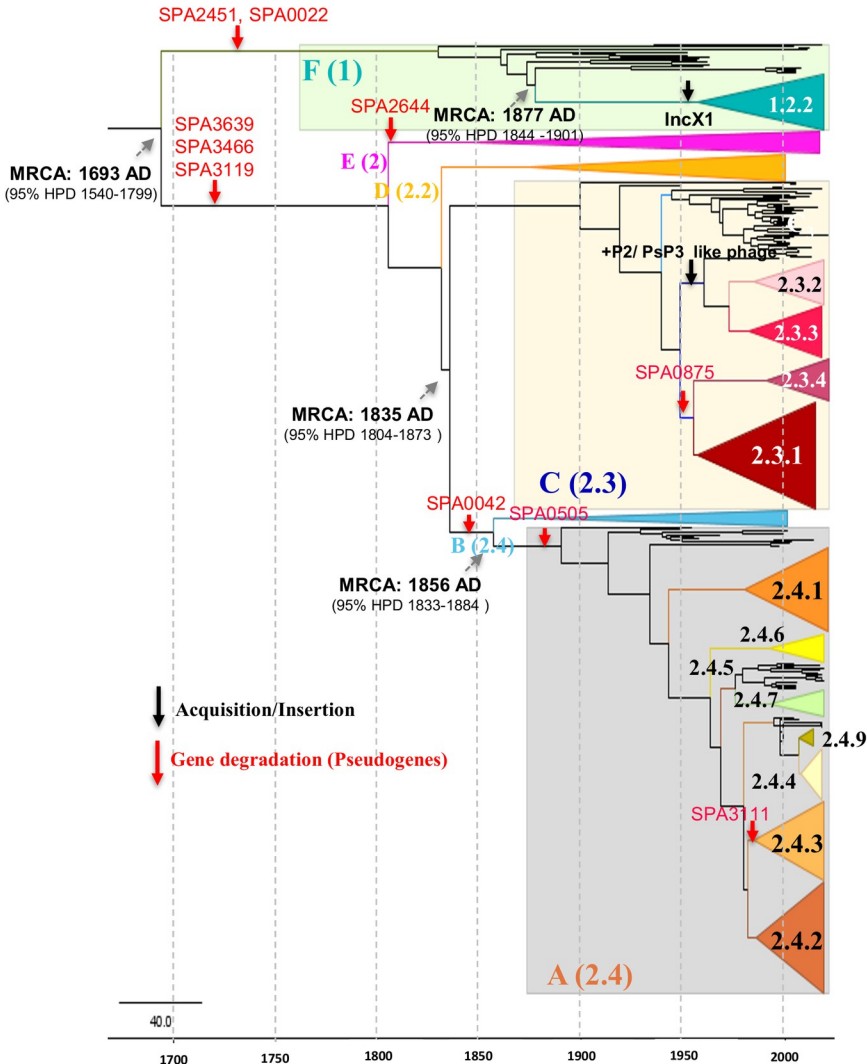

**Fig 2. Time-calibrated Bayesian phylogeny phylogenetic tree showing the evolutionary events (HDCS forming mutations, insertions and deletions) that define the seven modern lineages and sub-lineages of S. Paratyphi A.** Major lineages/ genotypes were simplified as colored cartoon triangles using FigTree (http://tree.bio.ed.ac.uk/software/figtree/). Red arrow represents frameshift mutation/ gene degradation, Black arrow represent acquisition/ gene gain. Grey arrows demarcate nodes of interest, and the accompanying data indicate 95% HPD of node heights.

all three lineages/sub-lineages in south Asian countries appear to be largely driven by the development of *gyrA* S83F mutation (except for a subcluster in 2.3 -*gyrA* S83Y). Though this mutation is not unique to these lineages, there is a strong association between reduced susceptibility to fluoroquinolones caused by the S83F mutation and the persistence/spread of these lineages. Our data is in line with the emergence of FQNS *S.* Typhi lineages with positively selected S83F mutants in South Asian countries [20]. Nevertheless, acquired AMR genes or mutations within these QRDR regions are not the sole factors that determine the evolution of *S.* Paratyphi A [18].

The evolution of *Salmonella* sp. is strongly associated with gene influx, genome degradation and rearrangement events that aid in host adaptation [21]. Modern isolates of *S.* Paratyphi A possess an average of 173 genome degradation events through pseudogene formation in comparison to the 25–35 pseudogenes observed in host generalists, such as *S.* Typhimurium [22].

Since *S.* Paratyphi A evolved into a human-specific systemic pathogen approximately 450 years ago, many of these adaptive mutations would have occurred very early [12]. The genetic features responsible for causing enteric fever were a perpetual change, while the recent micro-evolution is transient and will likely be removed by purifying selection in the future [12].

In our study, we also focused on critical events that may have contributed to the expansion or extinction of the seven modern lineages of *S.* Paratyphi A. Our observations indicate that the emergence of these lineages and sub-lineages was primarily associated with gene acquisitions or losses and mutations in genomic regions related to metabolism (**Fig 2**). Pan-genome analysis of SEFI isolates and representative isolates from a global collection showed the gain of prophages or plasmids during the selection of lineages (**Table 1**). Evaluation of gene degradation also depicted that disruption of metabolic pathways along the phylogenetic lineages/sublineages are key factors in evolution (**Table 2**). These findings further confirm that differences in metabolic functions due to environmental and/or human behavioral factors play a significant role in the expansion of lineages.

Identifying missense mutations occurring specifically in genes responsible for LPS biosynthesis is crucial since these genes are critical targets for developing vaccines and diagnostic assays [23]. Though the impact of these mutations on phenotype, fitness and evolution is currently unknown, the presence of lineage/genotype-specific association may be considered a signature of positive selection [24]. Among the six missense mutations, at least five have been predicted to stabilize the protein structure ($\Delta\Delta G \geq 0$). However, the experimental impact of these mutations will require more laboratory analyses. Further sequencing of isolates may reveal the existence of any selective pressure that may aid the genotypes in evading the host immune response. At present, *S.* Paratyphi A O-polysaccharide glycoconjugate vaccine is expected to have a protective response against all currently circulating *S.* Paratyphi A lineages.

Considering the limited number of isolates sequenced, this study has certain limitations. We only sequenced a representative collection of 152 isolates that were collected during the study period. Secondly, several isolates belonging to the global collection could not be assigned to genotypes by Paratype, which would necessitate the sequencing of more *S.* Paratyphi A isolates from the region in the future. Finally, and outside the scope of our study, a functional study of the reported mutations should be performed to indisputably explain their role in evolution. Despite these limitations, we could robustly evaluate the global phylogenomics of this mostly neglected pathogen with the collection we had. More extensive studies and continuous surveillance is needed to draw better public health policies for *S.* Paratyphi A control.

## Materials and methods

### Ethical statement

This study was approved by the Institutional Review Board (IRB) of Christian Medical College, Vellore (IRB Min No: 10393 dated 30.11.2016). Study participants were informed of the purpose and objectives of the study prior to the study. Written informed consent is obtained from every participant in the study before using their data for further research purposes.

**Study settings.** A total of 19 centres across the country, with a diverse and vast population, in a three-tiered surveillance system consisting of community-level health care settings (Tier 1), secondary hospitals (Tier 2) and tertiary care hospitals (Tier 3) were selected to form an Indian Typhoid network entitled "Surveillance of Enteric Fever in India" (SEFI) [25]. Subject selection and patient recruitment were systematically conducted at all sites. Blood samples from the patients presenting with non-specific febrile illness with suspected bacteremia were taken for culture using the automated BacT/ALERT system (bioMérieux, France). For tier 3 sites, patients with blood culture-confirmed *S.* Typhi and *S.* Paratyphi A cases were recruited

and clinical details were extracted from the electronic records. Typhoid and Paratyphoid cases detected in tier 3 linked laboratories from patients who were not enrolled in the study were also included in the study. Details of the isolates, participation centers and respective epidemiological settings are provided in the supplementary material (**S1 Table**).

**Bacterial isolates and antimicrobial susceptibility testing.**   Clinical isolates of *S. Paratyphi A* isolated from blood and bone marrow cultures from the participating centres were received at the central reference laboratory at the Department of Clinical Microbiology, Christian Medical College, Vellore, India. These isolates were further identified and confirmed as *S. Paratyphi A* by standard biochemical and agglutination tests by the Kauffmann-White scheme [26]. Antimicrobial susceptibility testing was performed for commonly used agents such as ampicillin (10 μg), chloramphenicol (30 μg), co-trimoxazole (1.25/23.75 μg), ciprofloxacin (5 μg), pefloxacin (5 μg), ceftriaxone (30 μg) and azithromycin (15 μg) by disk diffusion. Test results were interpreted as per clinical breakpoints recommended by the Clinical and Laboratory Standards Institute [27]. Azithromycin zone size interpretation was based on CLSI *S. Typhi* criteria (Sensitive $\geq$13 mm; Resistant $\leq$12 mm)

**Genomic DNA extraction and sequencing.**   A subset of 152 *S. Paratyphi A* isolates from the collection (*n = 152/441)* were selected for WGS by ensuring temporal and geographic representation across India. Each bacterial isolate was grown in LB broth (Oxoid) at 37˚C and growth was assessed by the increase in turbidity and by microbial count ($>10^9$ CFU/ml). The liquid cultures were centrifuged at 10,000 rpm and DNA was extracted from the pelleted cells using a Wizard DNA purification kit (Promega, Madison, USA) as per the manufacturer's protocol. The purity and concentration of extracted DNA were measured using Nanodrop One (Thermo Scientific) and Qubit dsDNA HS Assay Kit (Life Technologies).

Sequencing ready, paired-end library was prepared using 100 ng of DNA with the Nextera DNA sample preparation kit as per the manufacturer's instructions (Illumina, Inc., San Diego, USA). This was followed by sequencing on Illumina NextSeq 500 and HiSeq X 10 platforms with a paired-end run of 2X150 bp. Raw reads were quality-checked to remove adapters and the filtered high-quality reads were assembled using Unicycler (https://github.com/rrwick/Unicycler).

**Genome data acquisition and characterization.**   A global representation of *S. Paratyphi A* (*n = 400)* isolates was selected from a curated subset of Enterobase (http://enterobase.warwick.ac.uk/species/senterica/) and other previously published genomes [9,12,13,16–18]. The corresponding paired-end reads were downloaded from European Nucleotide Archive (ENA; http://www.ebi.ac.uk/ena). Genotypes were assigned from raw reads using Paratype (https://github.com/CHRF-Genomics/Paratype). The high coverage ($>50X$) reads were assembled using Unicycler v0.4.9 (https://github.com/rrwick/Unicycler). The assembled genomes were analyzed using Seqsero v2.0 [28] to confirm the antigenic profile of the serotype. Sequence types of the isolates were designated using the Multilocus sequence typing (MLST) pipeline available in the Center for Genomic Epidemiology (CGE) (https://cge.food.dtu.dk/services/MLST/). AMR genes, point mutations and plasmids were screened against Resfinder and PlasmidFinder databases by using ABRicate (https://github.com/tseemann/abricate). In total, 152 *S. Paratyphi A* study isolates from SEFI collection along with 400 genome sequences from the public database, were included. The complete list of genomes used in this study and metadata is available in **S2 Table.**

**Variant calling and Phylogenetic tree construction.**   The assembled genomes were mapped against the reference genome *S. Paratyphi A* ATCC 9150 (Accession No: CP000026.1) using Snippy v4.6.0 [29]. The core genome SNP differences between the genomes, with respect to the reference, were generated as an alignment file. The Maximum likelihood (ML) phylogenies were constructed using the Fasttree [30] with GTRGAMMA model and the generated

phylogenetic tree was visualized and annotated using iTOL [31]. Phylogenetic clusters were assigned using rhierBAPS [32] specifying two cluster levels with 30 initial clusters (snp.matrix, max.depth = 2, n.pops = 30, n.extra.rounds = Inf, quiet = TRUE).

To assess the temporal structure, root-to-tip genetic distances from (ML) tree against sample collection dates using TempEst v 1.5.1 (http://tree.bio.ed.ac.uk) was performed. Using the regression analysis of root-to-tip distances, an association between sampling times and genetic divergence (molecular clock) was determined. The timed evolution of *S*. Paratyphi A lineages was estimated using Bayesian phylogenetic methods available in BEAST v.1.10 [33,34]. The recombination-free alignment file was used as the input for the time-scaled phylogenetic analysis. The Hasegawa, Kishino and Yano model (HKY) substitution with different demographic models (Bayesian skyline, exponential and constant) was investigated. To determine the best-fitting coalescent model to describe changes in effective population size over time, log marginal likelihoods were calculated using path sampling and stepping-stone sampling methods. Finally, Bayes factor [35] was used to determine the best-fit model with the formula [logBF = logPr (D|M1)–logPr(D|M2)]. The selected Bayesian skyline with an uncorrelated relaxed clock model was run in 3 independent chains for 200 million with a sampling of 10000 generations. A burn-in of 20% was discarded from each run and resulting log files were combined using LogCombiner 1.8.1 [36]. The convergence and mixing were manually inspected using Tracer.v.1.7 [37] to ensure that all the parameters converged to an ESS of >200. The maximum clade credibility (MCC) tree was generated using Treeannotator v.1.8.2 [38]. The output was analyzed using Tracer v1.7, with uncertainty in parameter estimates reflected as the 95% highest probability density (HPD). The annotated phylogenetic tree was visualized using FigTree v.1.4.4 [39].

**Lineage wise mutation profiling.** Mutations were identified by *in-silico* determination of single nucleotide polymorphisms (SNPs) using the Snippy v4.6.0 mapping and variant calling pipeline (https://github.com/tseemann/snippy). To obtain SNPs, the draft genome of the study population was mapped against the annotated feature of reference genome *S*. Paratyphi A ATCC 9150 (CP000026.1). In-house written bash scripts were used to retrieve the pattern of mutation accumulation with respect to the phylogenetic lineages. Genes that contained either frameshift mutation or a premature stop codon were manually curated and classified as HDCS or pseudogenes. The identified HDCSin different lineages were compared with the data reported previously [13,22].

**Pan-genome analysis.** The pan-genome of all the study isolates of *S*. Paratyphi A (*n = 552*) was annotated using Prokka v. 1.14 [40] using a custom database created with "prokka-genbank_to_fasta_db" based on 1328 annotated *S*. Paratyphi A genomes downloaded from NCBI (https://www.ncbi.nlm.nih.gov/genome/browse/#!/prokaryotes/152/). To remove redundancy, CD-HIT version 4.8.1 was used with the following parameters: -T 0 -M 0 -g 1 -s 0.8 -c 0.90 [41]. The Prokka-compatible protein sequence fasta file (custom database) was confirmed to be used by the Prokka with relevant flags as follows—genus spa—usegenus—rfam—evalue 1e-05—coverage 50 (https://github.com/tseemann/prokka). The annotated draft assemblies in GFF3 format were used as input to evaluate pan-genome diversity using Panaroo [42]. Panaroo was run using its "strict" mode with 'remove invalid genes enabled -I option *.gff -o results—clean-mode strict—remove-invalid-genes—core_threshold 0.98 -t 6 -c 0.80. The gene presence or absence in each genome obtained was grouped according to the phylogenetic lineages (A-G) using twilight scripts (https://github.com/ghoresh11/twilight) with default parameters [43]. Gene gain or loss was curated manually and mapped into the timed Bayesian phylogenetic tree generated using Figtree (http://tree.bio.ed.ac.uk/software/figtree/).

## Mutations in LPS biosynthesis genes

Snippy based variant calling was performed on the assembled genomes (*n = 551*) using the *rfb* loci of strain ATCC 9150 (CP000026: 860063–884690) as the reference. SNPs and Indels occurring within the coding region of *rfb* loci were considered and the mutations were screened and arranged according to phylogenetic lineage in tabulated format. Whole-genome alignment (.full.aln) from the snippy output was used to build a maximum likelihood phylogeny using FastTree [31] with GTRGAMMA model. The generated phylogenetic tree was visualized and annotated using iTOL. The three-dimensional structures of rfb genes were modelled using ModWeb (https://modbase.compbio.ucsf.edu/modweb/) homology-based method. The quality of the model was evaluated using Ramachandran plot and the effect of mutations at a molecular level were computed as 'ΔΔG' on all subject structures, using FoldX version 4 (http://foldxsuite.crg.eu/node/196).

## Supporting information

**S1 Fig. Map of India showing the regional diversity of *S. Paratyphi A* genotypes.** Pie chart colours indicate the propotion of genotypes prevalent in three major geographical locations in India. Study sites are represented as per the settings. Color keys for all the variables are given in the inset legend. Shape file of India containing the boundaries of country, state and UT used with permission from GitHub repository (https://github.com/AnujTiwari/India-State-and-Country-Shapefile-Updated-Jan-2020) under a CC BY license. The base map was created using ArcGIS software by Esri (www.esri.com). ArcGIS and ArcMap are the intellectual property of Esri and are used herein under license. Annotations were added to the base map using Microsoft Powerpoint.
(TIF)

**S2 Fig. Rooted maximum likelihood phylogenetic tree of *S. Paratyphi A* isolates showing the comparative phylogenetic clustering by lineages, predefined sub-lineages, RhierBAPS population clustering (level 1) and Paratype genotyping scheme.** Lineages are represented by various colored branches. Sublineages, BAPS cluster and Paratype scheme are labeled as color strips.
(TIF)

**S3 Fig. Visualization of pan-genome analysis data by Panaroo of 552 *S. Paratyphi A* genomes.** (a) Pie chart indicates the core, soft core, shell and cloud genome composition of *S. Paratyphi A* genomes (b) Maximum likelihood tree of *S. Paratyphi A* genomes were compared to a matrix with the presence (blue) and absence (white) of the accessory genes found in the pan-genome. The image was prepared using Phandango (https://jameshadfield.github.io/phandango/#/)
(TIF)

**S4 Fig. Linear representation of acquired prophage regions (P2/ PSP3 phage) generated using Easyfig (https://mjsull.github.io/Easyfig/).**
(TIF)

**S5 Fig. Rooted maximum likelihood phylogenetic tree of *rfb* loci of *S. Paratyphi A* isolates derived from the whole genome alignment by mapping against the reference genome of *S. Paratyphi ATCC 9150 (Accession No: CP000026.1) using Snippy.** Lineages and genotypes are labelled as colour strips. Amino acid substitutions in the *rfb* loci are represented by heat maps.
(TIF)

**S1 Table. List of whole genome sequenced isolates collected from the participating sites of the SEFI network.**
(XLSX)

**S2 Table. List of *S*. Paratyphi A genomes used in this study with accession IDs and metadata.**
(XLSX)

**S3 Table. Distribution of *S*. Typhi and *S*. Paratyphi A isolates collected across the participating sites of the SEFI network.**
(XLSX)

**S4 Table. Antimicrobial susceptibility profile of *S*. Paratyphi A tested in the present study.**
(XLSX)

**S5 Table. Lineage-defining Frameshift mutations/stop codons in *S*. Paratyphi A genomes.**
(XLSX)

**S6 Table. Lineage-defining missense mutations in *S*. Paratyphi A genomes.**
(XLSX)

**S7 Table. List of lineage-defining mutations in the O-antigen biosynthesis genes (*rfb* region) of *S*. Paratyphi A and their predicted impact on protein structures.**
(XLSX)

## Acknowledgments

We thank Prof. Nicholas Grassly, Imperial College London, for assistance with study design and research proposal development. We gratefully acknowledge Dr. Arif M. Tanmoy, Dr. Senjuti Saha and Dr. Yogesh Hooda (CHRF, Dhaka, Bangladesh) for help with the genotyping analysis. We acknowledge Dr. Duncan Steele, Ms. Megan Carey & Dr. Supriya Kumar, Bill & Melinda Gates Foundation for their technical support throughout the study on behalf of SEFI consortium. We thank all the lab members involved in SEFI reference lab activities, especially Dr. Anushree Amladi, Ms. Baby Abirami S, Ms. Dhanabhagyam K, Ms. Beebi E, Ms. Suganya S, Ms. Udaya and Mr. Ayyanraj N, CMC Vellore implicated in phenotypic testing and stock culture maintenance. We would also like to thank all the members of SEFI consortium, Wellcome Trust Research Laboratory, CMC Vellore and core sequencing teams at the Wellcome Trust Sanger Institute for their contribution to genome sequencing. The authors thank Ms Catherine Trueman (Clinical Pharmacist, CMC Vellore) for helping with language editing.

## Author Contributions

**Conceptualization:** Shalini Anandan, Gordon Dougan, Jacob John, Gagandeep Kang, Balaji Veeraraghavan, Ankur Mutreja.

**Data curation:** Jobin John Jacob, Agila K Pragasam, Karthick Vasudevan, Tharani Priya Thirumoorthy.

**Formal analysis:** Jobin John Jacob, Agila K Pragasam, Karthick Vasudevan.

**Funding acquisition:** Gagandeep Kang, Balaji Veeraraghavan, Ankur Mutreja.

**Investigation:** Jobin John Jacob, Agila K Pragasam.

**Methodology:** Jobin John Jacob, Agila K Pragasam, Karthick Vasudevan, Ashita Singh, Balaji Veeraraghavan.

**Project administration:** Jacob John.

**Resources:** Pallab Ray, Madhu Gupta, Arti Kapil, Sulochana Putil Bai, Savitha Nagaraj, Karnika Saigal, Temsunaro Rongsen Chandola, Maria Thomas, Ashish Bavdekar, Sheena Evelyn Ebenezer, Jayanthi Shastri, Anuradha De, Shantha Dutta, Anna P. Alexander, Roshine Mary Koshy, Dasaratha R. Jinka, Sunil Kumar Srivastava, Jacob John, Gagandeep Kang.

**Software:** Karthick Vasudevan, Aravind Velmurugan, Monisha Priya Teekaraman.

**Supervision:** Shalini Anandan, Gordon Dougan, Jacob John, Gagandeep Kang, Balaji Veeraraghavan, Ankur Mutreja.

**Validation:** Jobin John Jacob, Agila K Pragasam, Balaji Veeraraghavan, Ankur Mutreja.

**Visualization:** Jobin John Jacob, Agila K Pragasam, Karthick Vasudevan, Aravind Velmurugan.

**Writing – original draft:** Jobin John Jacob, Agila K Pragasam, Karthick Vasudevan.

**Writing – review & editing:** Shalini Anandan, Gordon Dougan, Jacob John, Gagandeep Kang, Balaji Veeraraghavan, Ankur Mutreja.

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
