## [Decision Letter · Decision Letter 0]

2 Aug 2022

Dear Dr Mutreja,

Thank you very much for submitting your manuscript "Genomic analysis unveils the role of genome degradation events and gene flux in the emergence and persistence of S. Paratyphi A lineages" for consideration at PLOS Pathogens. As with all papers reviewed by the journal, your manuscript was reviewed by members of the editorial board and by several independent reviewers. In light of the reviews (below this email), we would like to invite the resubmission of a significantly-revised version that takes into account the reviewers' comments.

The reviewers were positive about your findings. However, reviewer 3 has identified some major issues that should be addressed in your revised manuscript.

We cannot make any decision about publication until we have seen the revised manuscript and your response to the reviewers' comments. Your revised manuscript is also likely to be sent to reviewers for further evaluation.

Sincerely,

Denise M. Monack

Section Editor

PLOS Pathogens

Denise Monack

Section Editor

PLOS Pathogens

Kasturi Haldar

Editor-in-Chief

PLOS Pathogens

orcid.org/0000-0001-5065-158X

Michael Malim

Editor-in-Chief

PLOS Pathogens

orcid.org/0000-0002-7699-2064

The reviewers were positive about your findings. However, reviewer 3 has identified some major issues that should be addressed in your revised manuscript.

Reviewer's Responses to Questions

**Part I - Summary**

Reviewer #1: This is an important study looking at the genomic epidemiology and population structure of a relatively neglected pathogen in India. Genome sequencing and appropriate bioinformatic analyses have been used. I'm very happy with the methodology and interpretation of results. The work is very significant in two main areas. Firstly in the insights it gives us into evolution of this pathogen and secondly in its relevance to vaccine/diagnostic testing. Overall the study was well designed and performed. The level of scholarship is what I would expect to see in PLOS Pathogens.

Reviewer #2: Jacob et al describe the population structure, distribution of genotypes within India and sequence polymorphisms associated with potential pseudogenes and AMR in 152 new S. Paratyphi A sequences from India in the context of 400 global isolates. The sequence analysis has been carried out to a high standard and the report of the population structure of Paratyphi A from India will significantly increases the number of genome sequences in the public domain for this country and globally. The population structure has been described with a new higher resolution nomenclature called paratype that builds on the population structure described by Zhou et al, 2014. However, beyond a description of the distribution of genotypes in North, South or West India, little new insight into paratyphoid disease in this country is reported and the significance of their distribution is not discussed. Analysis of sequence polymorphisms appears only an incremental advance on that published previously by Zhou et al, 2014 and it not clear from the text if any new potential pseudogenes were identified by this analysis. The title of the manuscript suggests that the role of sequence polymorphisms leading to potential loss-of-function has been investigated but this is not the case. Although their association with certain lineages suggests that they may be important in the success of the genotype, it seems misleading to suggest that their role has been established. Functional analysis of sequence polymorphism was carried out on mis-sense mutations in several rfb genes involved in biosynthesis of long chain LPS, motivated by the fact that these may affect efficacy of potential vaccines targeting LPS antigens. Although computational predictions suggested that most of these would not negatively impact function, experimental confirmation was limited to agglutination with the O2 antigen, despite O12 being the immune dominant antigen for Paratyphi A LPS.

Reviewer #3: This paper describes the genome analysis of a collection of S. Paratyphi isolates, predominantly from India. The results describe: the use of a new typing scheme “Paratype” from Bangladesh to define new lineages (as per the Paratype publication); then confirms lineage specific AMR genes as being rare; defines the loss of SPI2 in lineage G (a single isolate); the gain of a PsP3-like phage within sub-groups of lineage 2; the presence of an IncX plasmid in a sub-group of lineage F; some lineage specific mutations with specific acquisition points on the Paratype tree and a statement that the mutations seen in the LPS biosynthetic pathway are unlikely to cause a problem for the vaccine because the organisms still agglutinate the O2 antisera which was used to identify the isolates as Paratyphi A.

**Part II – Major Issues: Key Experiments Required for Acceptance**

Reviewer #1: I have no major issues with this paper that need to be addressed.

Reviewer #2: (No Response)

Reviewer #3: Overall, the paper contains useful data and the analysis provides insights into the strains of S. Paratyphi A circulating in the area. However, the analysis does not seem, to this reviewer, to reveal much of interest beyond what is presented in the original Paratype pre-print or is already know about Paratyphi A. The PsP3 phage is interesting, but the analysis is cursory. I cannot see any obvious cargo genes in the sequence nor if all of the 2.3.2/3 lineage isolates have the phage. I tried to look in the Supplementary Table for the genotypes 2.3.2 and 2.3.3 and the phage to see if the lineage remained extant but there are many Paratype genotypes presented as just 2.3 and whilst the AMR genes and the plasmids are there the phage is not.

The treatment of pseudogenes is interesting but needs to be repeated with a more rigorous attention to the impact of each mutation. See https://academic.oup.com/nar/article/50/9/5158/6576363

The comment in the abstract "We also identified that the six characteristic missense mutations associated with the

lipopolysaccharide (LPS) biosynthesis genes of S . Paratyphi A confer only a low structural impact and would therefore have minimal impact on vaccine effectiveness." Is not supported - the argument that the agglutinating antibodies mean there is no structural difference is not a justification as if the antibodies had not agglutinated the bacterial isolate it would have been excluded from the study.

**Part III – Minor Issues: Editorial and Data Presentation Modifications**

Reviewer #1: I would like to see see a brief description of the sampling framework (geographical distribution, coverage of population) in the introduction or the results. I appreciate that this information is given in methods and can be obtained from other papers describing SEFI but it would help the reader to have this information earlier in the paper.

Line 102 (paragraph). Several means are provided in this paragraph with no indication of range or confidence. Please could the authors provide slightly more information. I appreciate that the details are available in supplementary data but 95% CIs (or ranges) would help indicate if these means a representative or where there is a lot of diversity among sampling sites.

Line 110: The authors write "The antimicrobial susceptibility test demonstrated that 100% of S. Paratyphi clinical isolates (n=441) were non-MDR". This would be better stated as "no . Paratyphi clinical isolates (n=441) were MDR"

Line 114: The authors write "Overall, Indian S. Paratyphi A isolates were found to be generally non-susceptible to ciprofloxacin, while they continue to be susceptible to first-line agents." The use of the word "generally" is not acceptable without providing some data. Please provide a proportion and include the denominator.

Reviewer #2: 1. The title should be changed to reflect the conclusions more accurately. The role of genome degradation in emergence and persistence of Paratyphi A lineages is not addressed in this work.

2. Throughout the authors use non-susceptible where it is common to use ‘resistant’. It is not clear if there is a justification for this distinction. This does make understanding the manuscript more challenging as it is easy to confuse with susceptible, but also seems inappropriate to describe single mutations in QRDR of gyrA as non-susceptible, as these only decrease susceptibility.

3. 61-63. The authors predict that mutations in rfb genes would have no impact on vaccination. This is not based on good evidence. Indeed, in lines 211-212 they acknowledge that this needs to be investigated.

4. 65-67. Please clarify how intervention by means of genomic surveillance could avoid S. Paratyphi A becoming a threat similar to Typhi? It seems that surveillance could monitor the emergence but not intervene.

5. 78-80. Please provide a time frame for this apparent increase. But the data from the India surveillance described in this manuscript would suggest less than 20% and this should be discussed as it disagrees with these previous publications. Was there likely bias in the sampling of this or previous studies that could account for the different conclusions?

6. 111-112. Are these drugs still first line antibiotics considering that treatment primarily targets Typhi that are largely resistant? Some discussion of this in the introduction would help interpret the data here.

7. 112. Why 'remained at nearly 98.9%'. Is there data from previous analyses as a comparator?

8. 136-137. Please clarify how geographical representation affects whether sub-lineage designations are appropriate. Genomic diversity alone seems to be the important criterion for lineage designation.

9. line 151-152. The relevance of reporting ST in this section is not clear. This might be better worked into the section describing Paratypes.

10. 153-154. Supplementary table 4 states that several isolates were resistant to azithromycin but this is not stated here.

11. 154-157. Please edit to remove repetition from within this section and with the results previously stated earlier.

12. 158-160. These single mutations normally result in a moderate decrease in susceptibility (see PMID: 16895934) which would question the use of the term non-susceptibility. Suggest decreased susceptibility or increased resistance.

13. 175-176. Suggest highlighting that lineage G was represented by a single strain. This is important as in general these types of observations are only of interest of the polymorphism is present in multiple strains suggesting it is not a transient event.

14. 186-188. What is the evidence that these 'pseudogenes' were associated with the emergence of lineages. Seems to be stated as fact without describing the data. Further, were these identified previously in Zhou et al, 2014 and is there evidence that these are not transient. In general, this section would benefit from more detail.

15. 193. Why was the lineage G strain excluded? Why are the predicted common ancestor different from that described previously by Zhou et al, 2014?

16. 196. Evolutionary rate could be DNA or protein, suggest using nucleotide substitution rate.

17. 202-204. Only rfbG is present in a large cluster of strains, the other mutations are sporadic and present in multiple isolates that are identical, consistent with transient mutations rather than Darwinian selection.

18. 199. Please check that all of these genes are specifically involved in O:2 antigen biosynthesis. I suspect that most are involved in long chain LPS, O2 is a from a branching paratose modification of the O12 backbone. Are any of these genes involved in that modification?

19. 207-208. Please specify the antigen serum used. From discussion and Methods this is O2. Were there any differences in agglutination with O12, the immune dominant antigen of Paratyphi A? Is serum agglutination sensitive enough to detect changes that could affect immunogenicity?

20. 214-216. This needs clarification, what does 'evolutionary changes that favor genetic diversity' mean? the intended meaning of 'Evolutionary changes' is not clear but normally means selection of beneficial traits that requires genetic diversity but how this 'favours genetic diversity' is not clear at all.

21. 229-230. Please explain why 'The phylogenetic positioning of contemporary S. Paratyphi A isolates in lineage F was unexpected'.

22. 233-235. Please explain how ‘The emergence of genotype 1.2.2 can be attributed to the acquisition of IncX1 plasmid’. This unequivocal conclusion needs to be supported by some evidence other than coincidence.

23. 262-264. This section needs a more nuanced discussion since Zhou et al 2014 concluded that there were many transient mutations leading to pseudogenes. Therefore, presumably there are potential examples of lineages and sublineages associated with these that that expanded for reasons other than Darwinian selection for the sequence polymorphism. How did you distinguish between these events? Are the ten pseudogenes listed novel, or were they reported in Zhou et al?

24. 365-367. How were HDCS and pseudogenes distinguished? There does not appear to have been any functional analysis of HDCSs to confirm that they are pseudogenes. Therefore, ‘HDCS’ would seem more appropriate throughout.

25. Discussion: What is known of the potential role of genes affected by potential loss-of-function mutations and how their loss could affect emergence and persistence?

Reviewer #3: Line 136 et seq: I find this confusing, are the authors saying that the Paratype scheme should be used?

Line 158 "Fluoroquinolone non-susceptibility in dominant lineages (A, C and F) of S. Paratyphi A was driven mainly by gyrA-S83F substitutions, with a few isolates harboring gyrA-S83Y (predominantly genotype 160 2.3) variant. Also, a significant number of isolates were fluoroquinolone susceptible with no mutations in the quinolone-resistance-determining region (QRDR), particularly genotype 2.3.1 (Figure 1)."

The prediction of MIC from QRDR sequence is known to be unreliable, E.g. https://www.nature.com/articles/s41598-020-64346-0/tables/3

Line 178 – “For example, the C4 sub-lineage (genotype 8 179 2.3.2 and 2.3.3) of S. Paratyphi A has acquired prophage regions P2/ PSP3 phage that could account for 180 their host specificities (Suppl Fig. 4)”

I do not follow this argument - all SPA are host restricted whether they have the phage or not.

PLOS authors have the option to publish the peer review history of their article (what does this mean?). If published, this will include your full peer review and any attached files.

Reviewer #1: **Yes: **Mark A Holmes

Reviewer #2: No

Reviewer #3: No
---

## [Decision Letter · Decision Letter 1]

27 Jan 2023

Dear Dr Mutreja,

Thank you very much for submitting your manuscript "Genomic analysis unveils genome degradation events and gene flux in the emergence and persistence of S. Paratyphi A lineages" for consideration at PLOS Pathogens. As with all papers reviewed by the journal, your manuscript was reviewed by members of the editorial board and by several independent reviewers. The reviewers appreciated the attention to an important topic. Based on the reviews, we are likely to accept this manuscript for publication, providing that you modify the manuscript according to the review recommendations.

Please address Reviewer #1's suggestion for adding confidence intervals.

Sincerely,

Denise M. Monack

Guest Editor

PLOS Pathogens

Kasturi Haldar

Editor-in-Chief

PLOS Pathogens

orcid.org/0000-0001-5065-158X

Michael Malim

Editor-in-Chief

PLOS Pathogens

orcid.org/0000-0002-7699-2064

Please address Reviewer #1's suggestion for adding confidence intervals.

Reviewer Comments (if any, and for reference):

Reviewer's Responses to Questions

**Part I - Summary**

Reviewer #1: As for submitted version:

This is an important study looking at the genomic epidemiology and population structure of a relatively neglected pathogen in India. Genome sequencing and appropriate bioinformatic analyses have been used. I'm very happy with the methodology and interpretation of results. The work is very significant in two main areas. Firstly in the insights it gives us into evolution of this pathogen and secondly in its relevance to vaccine/diagnostic testing. Overall the study was well designed and performed. The level of scholarship is what I would expect to see in PLOS Pathogens.

Reviewer #2: The authors have addressed all relevant comments through edits or removing text that was not supported by the data or analysis. The manuscript now describes additional genome sequences of S. Paratyphi A and an improved higher resolution analysis of the population structure that represents a significant yet limited advance on that published previously.

Reviewer #3: All corrections have been addressed

**Part II – Major Issues: Key Experiments Required for Acceptance**

Reviewer #1: Nothing major but I would like to see confidence intervals (see below)

Reviewer #2: No major issues need addressing

Reviewer #3: (No Response)

**Part III – Minor Issues: Editorial and Data Presentation Modifications**

Reviewer #1: I'm happy with the author's responses other than their reluctance to provide confidence intervals for their proportions. They do not seem to appreciate that if they repeated their study a hundred times they wouldn't always get the same result for their proportion. A 95% confidence interval will provide a range within which the true proportion will be found in 95 out of 100 repeats of the study. If they have difficulty in doing the calculation an online calculator can be found here:https://sample-size.net/confidence-interval-proportion/

It really isn't difficult to calculate and it is important to acknowledge some degree of uncertainty.

Reviewer #2: No minor issues need addressing

Reviewer #3: (No Response)

PLOS authors have the option to publish the peer review history of their article (what does this mean?). If published, this will include your full peer review and any attached files.

Reviewer #1: No

Reviewer #2: No

Reviewer #3: **Yes: **John Wain

Figure Files:

Data Requirements:

Reproducibility:

References:

---

## [Editor Report · Decision Letter 2]

27 Mar 2023

Dear Dr Mutreja,

We are pleased to inform you that your manuscript 'Genomic analysis unveils genome degradation events and gene flux in the emergence and persistence of S. Paratyphi A lineages' has been provisionally accepted for publication in PLOS Pathogens.

Best regards,

Debra E Bessen

Section Editor

PLOS Pathogens

Denise Monack

Guest Editor

PLOS Pathogens

Kasturi Haldar

Editor-in-Chief

PLOS Pathogens

orcid.org/0000-0001-5065-158X

Michael Malim

Editor-in-Chief

PLOS Pathogens

orcid.org/0000-0002-7699-2064
---

## [Editor Report · Acceptance letter]

25 Apr 2023

Dear Dr Mutreja,

We are delighted to inform you that your manuscript, " Genomic analysis unveils genome degradation events and gene flux in the emergence and persistence of S. Paratyphi A lineages ," has been formally accepted for publication in PLOS Pathogens.

Best regards,

Kasturi Haldar

Editor-in-Chief

PLOS Pathogens

orcid.org/0000-0001-5065-158X

Michael Malim

Editor-in-Chief

PLOS Pathogens

orcid.org/0000-0002-7699-2064